# The Role of Insects in Sustainable Animal Feed Production for Environmentally Friendly Agriculture: A Review

**DOI:** 10.3390/ani14071009

**Published:** 2024-03-26

**Authors:** Csaba Hancz, Sadia Sultana, Zoltán Nagy, Janka Biró

**Affiliations:** 1Kaposvár Campus, Hungarian University of Agriculture and Life Sciences, Guba S. 40., 7400 Kaposvár, Hungarysadiazoology@nstu.edu.bd (S.S.); 2Research Center for Fisheries and Aquaculture, Institute of Aquaculture and Environmental Safety, Hungarian University of Agriculture and Life Sciences, Anna-liget 35., 5540 Szarvas, Hungary

**Keywords:** insects, sustainability, ecosystem services, insect products

## Abstract

**Simple Summary:**

This paper provides an overview of the key issues concerning the role of edible insects in sustainable feed production and environmentally friendly agriculture. The use of insect-derived feeds for animal production is presented in detail and discussed for the major terrestrial livestock and aquaculture groups.

**Abstract:**

The growing demand for animal protein, the efficient use of land and water, and the limitations of non-renewable energy sources highlight the global importance of edible insects. This paper provides an overview of the key issues regarding the role of edible insects in sustainable feed production and environmentally friendly agriculture. The indispensable ecological services provided by insects are discussed, as well as the farming, products, and nutritional value of edible insects. A representative selection of the literature reviewing major insect species’ chemical compositions and nutritional value is also presented. The use of insect-derived feeds for animal production is presented in detail and discussed for the major terrestrial livestock and aquaculture groups.

## 1. Introduction

Sustainability, circularity, and environmental friendliness are widely acknowledged to be essential components of contemporary human endeavors. Simultaneously, food production for a rapidly expanding global population remains a pressing concern in the face of scarce opportunities for expanding arable land. To ensure high levels of sustainable agricultural production, it is crucial to incorporate all available knowledge. Thus, the significance of insects must be re-evaluated in this context. Dunkel and Payne [1] offer a comprehensive overview of the global significance of edible insects in their introduction to “Insects as Sustainable Food Ingredients,” highlighting the growing demand for animal-based protein, the efficient use of land and water, and the limitations of non-renewable energy sources. Highlighting the increasing concern for sustainability, Guiné et al. [2] present important findings on the efficiency of insects in comparison to other farm animals. They note that the environmental impact of insect production encompasses feed conversion, land use, and water consumption. Insects require the least amount of feed, land, and water, followed by chickens, pigs, and cows. In contrast, cows require over five times as much feed, land, and water as insects. Calculations for protein efficiency, expressed in terms of protein concentration, reveal that beef, pork, and poultry have protein values of 190 g/kg, 50 g/kg, and 200 g/kg, respectively, while crickets have values of 154 g/kg and 205 g/kg for nymphs and adults, respectively [2].

Insects represent the largest and most diverse group of animals, with over a million described species, accounting for more than half of all known living organisms, i.e., over 90% of the animal life forms on Earth. The roles of insects are fundamental and diverse. They play a crucial role in providing ecosystem services, including pollination, biological control, the provision of food, and organic matter recycling. Insects represent a dominant component of biodiversity across most terrestrial ecosystems and are critical to nutrient cycling and overall ecosystem functioning [3,4]. Although herbivorous insects cause damage to about 18% of global agricultural production, fewer than 0.5% of all known insect species are considered pests [5]. Aquatic insects are also highly diverse, with over 100,000 species inhabiting freshwater ecosystems, and they play crucial ecological roles, such as decomposing organic matter and transferring energy between trophic levels. These hexapods serve as links in the nutrient cycle, and their biological interactions have substantial impacts on the community structure within the freshwater ecosystem [6].

Insects possess the potential to tackle contemporary global challenges and advance the Sustainable Development Goals (SDGs) owing to their significant influence on various ecosystem services [7]. Ecosystem services are frequently undervalued; hence, a lucid illustration of this, in the context of insects, would be valuable (see Figure 1). However, Schowalter et al. [8] introduced the concept of ecosystem disservices, which serves as an important counterpoint to insect ecosystem services and warrants due consideration. Insect larvae can provide atypical ecosystem services, such as plastic degradation and biofuel production, which have the potential to aid in mitigating the effects of climate change, as suggested by Morimoto and Harris [9], and could become a priority in the future.

In addition to ecosystem services, another fundamental benefit of insects is their mass production based on agricultural and other organic wastes. Insects efficiently bioconvert organic waste into new products, mirroring the principle of a circular economy, providing benefits such as reduced waste management costs and resource use in protein and fat production through insect-derived feed [11].

Insects have served as food for humans since the earliest times, and their consumption (entomophagy) is included in traditions in many areas of the world. It is impossible to find reliable data on the amount of direct human consumption, but currently, according to Govorushko [12], 2 billion people in 113 countries across Africa, Asia, Central and South America, and Australia consume insects, although Western attitudes towards this type of food are certainly ambiguous. Liceaga et al. [13] provide a nicely illustrated review of the current situation regarding the potential use of insects and insect products for human consumption.

However, in their stimulating and thorough review, Akhtar and Isman [14] emphasize that in situations where there is an inadequate supply of conventional animal protein sources such as beef, pork, and chicken, alternative sources such as insects need to be found. Edible insects have great potential as an environmentally friendly choice for future food systems, primarily because of their high nutritional content. In addition, insects have greater food conversion efficiency and produce lower levels of greenhouse gas emissions (GHGEs) while requiring less water and land than vertebrates, a serious consideration for present and future generations.

A decade ago, the FAO called for increased use of forage insects in feed production to conserve the environment and resources and ensure food and feed security in the face of a growing world population [15]. An overview of recent developments in the use of insects as food and feed was provided by van Huis [16], who reviewed the literature on edible insects in recent years and noted that attention shifted from insect harvesting to insect farming about a decade ago. In addition, most production is for pet food but will soon turn to aquafeed. (As discussed below, other major animal production sectors are also increasingly using insect meal in feeds.) In addition to the obvious economic and environmental considerations, the insect food and feed sector is developing rapidly thanks to an increasingly supportive regulatory framework. This latter aspect has recently been highlighted by Sogari et al. [17], who, among other things, summarized the pertinent EU regulatory framework, according to which insects for human consumption fall under the category of so-called novel foods (NFs) (as of 2018). Chia et al. [18] discuss another important aspect of insect farming: it has the potential to promote inclusive business opportunities for smallholder farmers within the agribusiness value chain while also contributing to solving socio-economic and environmental problems in developing countries in ways that align with the United Nations’ Sustainable Development Goals.

Finally, this section provides an overview of worldwide market statistics and expectations related to edible insects. According to market research, the Edible Insects Market is projected to reach USD 9.60 billion by 2030, with a Compound Annual Growth Rate (CAGR) of 28.3% during the forecast period of 2022–2030. In terms of volume, the market is expected to reach 3,139,035.10 tonnes by 2030, with a CAGR of 31.1% during the forecast period 2022–2030. The edible insect market is growing due to increased greenhouse gas emissions from the livestock and poultry industries, the high nutritional value insects offer, the environmental advantages of eating them, the rising demand for insect-derived protein in animal feed, and the low risk of acquiring zoonotic diseases when consuming them. Based on products, the whole segment is projected to make up the greatest share of the edible insect market in 2022. This is primarily due to the readily available and lower cost of whole insects compared to processed insects. Additionally, the lack of insect-processing facilities in certain regions of the world and the increasing demand for insects in the animal feed industry contribute to the growth of the whole insect segment. Regarding the types of insect, black soldier fly production is expected to experience significant growth during the 2020–2030 forecast period. This growth is being driven by the expanding aquaculture industry, rising demand for alternative proteins in the animal feed sector due to increased soy meal prices, government backing of insect meal use in livestock feed, and increased investment by major players in the black soldier fly industry. Based on its end use, the segment of human consumption is anticipated to rapidly grow during the 2020–2030 forecast period. This growth is attributed to several factors, including the increasing demand for insect-based foods to feed a growing global population, the high nutritional value insects can provide to humans, and the rising demand for environmentally friendly protein alternatives. Based on geography, the Asia–Pacific region is expected to account for the biggest portion of the edible insect market in 2022 [19].

## 2. The Role of Insects in Agriculture

Intensified agricultural production systems use huge external inputs to increase yields, but they also destabilize ecosystems [20]. Biodiversity is the key to sustainable agricultural production and, more broadly, a livable planet. Understanding the role of insects in ecosystems will help in achieving this goal. As briefly mentioned above, insects also play an important role in agroecology by acting as drivers of ecosystem functions. Ecologically sustainable management of agricultural systems includes the use of beneficial organisms [21]. According to Jankielsohn [5], the main functions of insects in the ecological processes of terrestrial ecosystems are nutrient cycling, seed dispersal, bioturbation [22,23], pollination [24,25], and biological pest control [26,27]. Cock et al. [28] group important invertebrates in agricultural ecosystems into soil invertebrates, biological control agents (BCAs), and pollinators. Follett et al. [29] provide a new perspective on BCAs, including with respect to environmentally friendly methods such as intercropping and insect habitat modification. In a review of mass-produced insects for biological control, most of which are used for agricultural purposes, Francuski and Beukeboom [30] list 33 species. Also worth mentioning is their subcategory of sterile insect technology, which includes nine species.

As the mass rearing of insects for food and feed is undergoing full industrial development as an effective alternative to animal protein production, its by-product, insect excreta (frass), must also be considered for use as an organic fertilizer to replace the use of artificial ones, contributing to sustainable agriculture and a circular economy [31]. Insect frass and exuviae, which are mostly composed of chitin, exert beneficial effects in a variety of ways, such as on plant growth, resistance, and reproduction, or by stimulating the soil microbial community [32]. For these reasons, they could certainly be a valuable alternative to conventional fertilizers and pesticides [17].

## 3. Edible Insects, Farming, Products, and Nutritional Value

Francuski and Beukeboom [30] provided an excellent review of the literature on insect mass production. They list a total of 62 species in the following categories: biological control (33), food (6), feed (5), pollination (4), industrial production (3), medicine and cosmetics (4), waste management (2), and research (5). According to Nolan et al. [33], about 2300 edible insects have been reported, belonging to a large number of different phyla and species and having diverse sets of morphological characteristics, showing great variation in composition and nutritional aspects. Factors affecting the digestibility and bioavailability of the digested proteins were studied, and the effect of the digested edible insect protein on the gut microbiota was summarized in their work; additionally, a critical evaluation of the positive and negative aspects of edible insect protein consumption was given.

Cortes Ortiz et al. [34] provided an overview of all aspects of insect mass production, covering its history, which began with pest control objectives in the 1970s; the species used and their respective applications; production techniques; and environmental control. Like many others, they emphasize that developments in insect mass production will be directed toward food and feed production. The benefits of insect production and its current status in Europe are well illustrated by Thrastardottir et al. [35], whose review focuses on the most popular insects farmed in Europe, the yellow mealworm, *Tenebrio molitor*, and the black soldier fly (BSF), *Hermetia illucens*, together with the main obstacles and risks. According to Hong et al. [36], *T. molitor* larvae are being mass-produced as feed for pets, zoo animals, and production animals. These larvae are a highly sustainable protein source, containing high-quality protein present in high quantities, and can replace soybean meal or fishmeal. Supplementation with *T. molitor* larvae improved the growth performance of broiler chickens without having negative effects on traits [37]. In studies involving swine, supplementation of *T. molitor* larvae improved performance and protein utilization for growing pigs [38].

The main species involved in mass production worldwide are listed in Table 1.

For the time being, progress has reached a level where the effect of strain on the overall performance of yellow mealworms (*T. molitor*) has been studied. Rumbos et al. [39] found that an Italian strain had the best growth performance in terms of survival, larval biomass production, development time, and feed utilization. The results of their study highlight the need to select strains that can increase the overall productivity of insect farming systems.

**Table 1 animals-14-01009-t001:** Major insect species farmed for food and/or feed.

Scientific Name	Common Name	Developmental Stage	Food	Feed
*Acheta domesticus*	House cricket	Adult	x	x
*Tenebrio molitor*	Mealworm	Larvae	x	x
*Gryllus bimaculatus*	Mediterranean field cricket	Adult		x
*Bombyx mori*	Silkworm	Larvae, pupae	x	x
*Galleria mellonella*	Wax worm	Larvae	x	
*Apis mellifera*	European honeybee	Adult	x	
*Musca domestica*	Common housefly	Larvae		x
*Lucilia sericata*	Common green bottle fly	Larvae (maggot)		x
*Rhynchophorus ferrugineus*	Red palm weevil	Larvae, pupae	x	
*Rhynchophorus phoenicis*	Palm weevil	Larvae	x	
*Pachnoda marginata*	Sun beetle	Larvae	x	x
*Hermetia illucens*	Black soldier fly (BSF)	Larvae	x	x

Sources: [34,40].

The metabolic flexibility and efficiency of insects are certainly the most important aspects of mass production for food and feed. Ramos-Elorduy [41] emphasizes that this is mainly based on the well-known fact that insects are poikilothermic. According to her data, the energy content of edible insects varies between species and regions, but in general, Coleoptera and Lepidoptera species provide more energy. The energy values for livestock are 165–705 kcal/100 g, and those for vegetables 308–352 kcal/100 g, while edible insects provide 217–777 kcal/100 g, and insects raised on organic waste provide 288–575 kcal/100 g. Waldbauer [42] provided perhaps the first comprehensive review of insect food consumption and utilization, analyzing differences among species, diets, and environmental factors. According to Maino and Kearney [43], insects are characterized by their small size, large numbers, impressive reproductive output, and rapid growth that follows a qualitatively different trajectory than that of many other animals. Their model of insect growth predicts that energy reserves per biomass increase with age, implying higher production efficiency and biomass energy density in later instars. This model also suggests that insects achieve greater production efficiency and higher growth rates by increasing specific assimilation and increasing energy reserves per biomass, which are less costly to maintain than structural biomass. A long path of research and development led to the study of Halloran et al. [44], who were already able to analyze the environmental impacts of different insect production systems based on life cycle inventories. In addition, the need for some genetic work to increase the productivity of insect farming systems has also emerged to take advantage of the given metabolic characteristics [39].

There is an abundance of research discussing the chemical composition and nutritional value of insects. Weru et al. [45], in their review on edible insects as a suitable source for human nutrition, screened hundreds of articles. Their data showed a large variation in nutritional value, even within species, according to sex, geographical origin, and growth stage. It has to be mentioned that the quality of much of the published data was poor because of the small number of samples analyzed.

Table 2 contains mean data on crude protein and crude fat content, constituting the main characteristics of the nutritional value of some edible insects. Some references are given where the standard deviation shows rather high values, demonstrating that the chemical composition of insects is highly variable due to several factors, such as species, diet, sex, developmental stage, and processing method.

The nutritional value of insect meal cannot be assessed without its amino acid profile and digestibility, constituting exactly what can be found in the Makkar et al.’s [48] paper, which also examined the latter according to consumer species. According to Sánchez-Muros et al. [51], who provided a long list of the chemical compositions of species, insects can have an adequate amino acid profile, depending on the insect species. The most common limiting amino acids are histidine, lysine, and tryptophan, which can be incorporated into the diet.

The other component of the insect body to consider is chitin. Chitin has both beneficial functions (e.g., prebiotic effects) and putative negative effects on digestibility [47]. However, chitin is a relatively small component of most insects; e.g., Sándor et al. [50] found its proportions to be 9.6 and 5.5% in dry matter in BSF and mealworm meal, respectively. On the other hand, some animal species have chitinases in their gastrointestinal tracts. Where this is not the case, the inclusion of insect meal in feed must certainly be limited [52].

In addition to nutritional value, insects also contain bioactive compounds that serve as health stimulators in livestock animals. Animal researchers regularly focus on three categories of bioactive compounds present in insects: antimicrobial peptides (AMPs), fatty acids (especially lauric acid), and polysaccharides such as chitin and chitosan. Each of them demonstrates antimicrobial activity through various mechanisms, such as forming or destroying membrane pores, impeding intracellular processes, or supporting the host’s immune system. Additionally, the antioxidant capabilities of insect proteins may protect against oxidative tissue damage. The combination of AMPs, which destroy the bacterial cell envelope with fatty acids, and chitin or chitosan may provide a solution to antimicrobial resistance [53].

## 4. Use of Insect-Derived Feeds in Animal Production

Insect meal is considered a viable solution to mitigating the limitations imposed by natural resource depletion, climate change, and the consumption rivalry between food, feed, and fuel. Due to their increasing demand and resulting cost escalation, primary protein sources for animal nutrition such as soybeans, peas, and fish meal are no longer sustainable for long-term use. Insects like black soldier fly larvae, crickets, and mealworms can effectively supplement feed sources by offering valuable energy, protein, and fat for an animal’s diet [54].

The costs of conventional feed resources such as soybean meal and fishmeal are very high, and their availability will be limited in the future. Insect farming could therefore be part of the solution. The nutritional quality of black soldier fly larvae, house fly maggots, mealworms, locusts/grasshoppers/crickets, and silkworm meal and their use as replacements for soybean meal and fishmeal in the diets of poultry, pigs, fish species, and ruminants were discussed by Makkar et al. [48]. Their main findings are as follows: the protein content of these alternative resources is as high as 42 to 63%, and so is their lipid content, which can be extracted and used for various applications, even for biodiesel production. The concentration of unsaturated fatty acids is also high but varies between species. According to the studies discussed, the palatability of these alternative diets is good for the animals, and these resources can replace 25–100% of soybean meal or fishmeal, depending on the species. Digestibility analyses of dog food containing BSF larvae meal as the sole source of protein (36.5% inclusion) showed values that off meat-based diet, indicating its suitability as a sustainable protein source for pet food [55]. However, most types of insect meal are deficient in calcium, so its supplementation is necessary, especially for growing animals and laying hens. Gasco et al. [56], besides reviewing the factors affecting the decision-making process regarding using insect-based products, also provide a comprehensive summary of the field, ranging from the species involved to processing methods and questions of safety, cost, availability, and the consistency of supply and legislation. Important aspects of product quality such as digestibility, palatability, and even pelletability are also discussed.

Insect-derived feeds are part of many animal feeding programs because they provide a sustainable way of recycling waste into nutrient-rich ingredients. Secondly, since many animals naturally consume insects, their inclusion in feeding programs can actually improve animal welfare. In addition, the nutrient composition and availability of insect-derived ingredients are generally very high relative to requirements and formulation needs, and components such as chitin, proteins, and fatty acids with antimicrobial activity provide additional benefits, as Koutsos [57] summarized. Last but not least, serious economic considerations will also appear soon with regard to using them to replace other animal protein sources, especially fishmeal. However, due to low produced quantities and high production costs, the price of insect-based proteins is still high and not competitive when compared to that of fishmeal or soybean meal, although continued efforts to upscale production will increase product availability and quality/consistency and reduce costs [58]. However, the data provided by Pinotti et al. [59] deserve attention. They compared the prices of different insect meals with those of soybean and fish meal and found that the defatted meal of BSF and crickets were 3 and 7–12 times more expensive, but if the comparison was made on 100 g of protein base, these values were 2 and 6–9. Shah et al. [49] made a similar comparison based on a literature review that included housefly maggots, mealworms, and BSF. According to their evaluation, housefly maggots and mealworms showed very favorable values both concerning product price and that calculated for CP, while the values of BSF were much higher. However, it should be noted that these data for BSF came from only two review articles, so their accuracy may be questionable.

The main and most important insect-derived products are meal and oils, which are used as feed ingredients in increasing quantities and proportions for a growing number of species [60]. Although insects are a rich source of fat that could be very well used in animal feeds, facilitating even pelleting, this practice is not widespread due to this product’s short shelf life. Other beneficial effects of insect oil, namely, that of BSF larvae, were revealed by Prachumchai and Cherdthong [61], who found, in their study using in vitro gas production, that adding 4% black soldier fly larva oils (BSFOs) at different roughage (R)-to-concentrate (C) ratios increased propionate levels, decreased methane emission, and preserved dry matter (DM) degradability. In addition, insect oils have other applications, such as use in green cosmetics or biodiesel production [17].

## 5. The Use of Insect Meal in the Feeding of Terrestrial Farm Animals

### 5.1. Ruminants

While live insects and their products have been shown to be suitable sources of protein and fat that can be used in the diets of farmed monogastric animals, very little information is available on the effects of insect inclusion in ruminant diets on feed digestibility and performance. An important reason for this is that the use of insects in ruminant diets is currently banned in many countries worldwide due to the risk of BSE, although there is no evidence to date that insects carry and transmit prions [62]. Nevertheless, some in vitro studies have highlighted the relatively low digestibility of insects for ruminants, unless developing processing technologies can improve this aspect. Furthermore, in vivo feeding studies are needed to evaluate the impact of live insect products and by-products on the quantitative and qualitative aspects of ruminant production and health [63]. The main conclusion for the future, according to Renna et al. [64], is that a major research effort is needed in the coming years to achieve a thoughtful nutritional evaluation of insect-derived products (i.e., whole and defatted meals, oils, and other ingredients). There is also a need to establish specific recommended inclusion levels for ruminants, taking into account the wide variation in insect composition due to species and production systems. It would be advisable to evaluate the use of insect products in commercial ruminant diets, taking into account their economic and environmental impacts on different farming systems.

An example of the much-needed research in this area was published by Toral et al. [65], who compared the in vitro ruminal protein degradation of four insect species using three methods. They found that all the analyzed techniques appeared to provide similar rankings, with good correlations between methods, particularly between regression and in situ results. Irrespective of the methods, nitrogen from the four insects did not show high ruminal degradation (41–76%). However, among the species studied, *T. molitor* showed the lowest and highest values of ruminal N degradation and intestinal digestibility, respectively, so it may be the best option for replacing dietary soybean meal in ruminant diets.

### 5.2. Pigs

Although they primarily focus on the Australian context and BSF, as a protein and fat source, DiGiacomo and Leury’s [62] literature review provides valuable insights into the use of insect meal in pig feeds. Their review indicates that while fat content affects palatability and digestibility, BSF larvae constitute a suitable ingredient for pig diets, as demonstrated by previous studies [66]. Since the inclusion level was 33% in this early experiment with 5-week-old barrows, the observed decrease in dry matter digestibility can be understood, as well as the increase in feed intake. In a subsequent study [67], where early-weaned pigs were fed with 0%, 50%, and 100% replacement rates of dried plasma with black soldier fly larvae, it was found that 50% replacement increased while 100% decreased pigs’ performance. Replacing the soybean meal in growing pigs’ diets with 50, 75, and 100% partially defatted BSF meal (61% CP and 14% lipid) had no adverse effect on pork quality or sensory parameters. Furthermore, the inclusion of BSF larvae improved juiciness in the supplemented groups. Pigs fed with BSF larvae produced back fat with a higher polyunsaturated fatty acid content, likely due to the larvae’s high-fat content [68]. Biasato et al. [69] conducted an experiment where soybean meal was replaced with BSF larvae at rates of 0%, 30%, and 50% in the diets of weaned pigs. This study revealed a linear increase in daily feed intake; however, no effects on growth were observed. Jin et al. [38] replaced soybean meal with dried mealworms at lower rates, namely, 1.5%, 3%, 4.5%, and 6%, which caused a linear increase in growth and digestibility parameters in weaned female piglets.

Among edible insects, according to Hong et al. [70], the black soldier fly, yellow mealworm, and common housefly are viable protein source alternatives for pigs. Both can replace fish meal in diets for weaned pigs without having any adverse effects on growth performance or nutrient digestibility. Moreover, insect products exhibit higher standardized ileal digestibility values of amino acids compared to conventional animal proteins in growing pigs [70].

*T. molitor* larvae have been industrially produced as feed for pets, zoo animals, and even production animals. They represent a protein-rich alternative to soybean meal or fishmeal, boasting a high quality and quantity of protein and an optimal amino acid profile. Therefore, they are considered an environmentally sustainable protein source. In swine studies, supplementing mealworm larvae enhanced growth performance and protein utilization for weaning pigs. Additionally, growing pigs exhibited greater amino acid digestibility when fed 10% *T. molitor* larvae than when they were fed conventional animal proteins [70]. However, Hong et al. [70] suggest that further studies are required on the optimal inclusion level of insect products in every phase of pig diets, from weaned pigs to sows. This is because previous studies have predominantly been conducted on weaned pigs and using a limited number of insect products.

### 5.3. Poultry

De Marco et al. [37] discovered how many of the nutrients in two types of insect larval meals (*T. molitor* and *H. illucens*) were digested by broiler chickens. They also figured out how much energy the chickens acquired from the meals. Finally, they measured the amounts of amino acids that were digested in the gut. The study tested three diets: a basic diet and two diets made by replacing 250 g/kg (*w/w*) of the basic diet with either *T. molitor* meal or *H. illucens* meal. The coefficients of total tract apparent digestibility (CTTAD) of the nutrients for both insect larval meals did not differ significantly, though differences were found for the CTTAD of ether extract for apparent metabolizable energy (AME) or nitrogen-corrected apparent metabolizable energy (AMEn) (AME = 16.86 and 17.38 MJ/kg DM, respectively; AMEn = 16.02 and 16.60 MJ/kg DM, respectively). This study concluded that both *T. molitor* and *H. illucens* meal can be viable sources of AME and easily digestible amino acids (AAs).

Schiavone et al. [71] found that using fat derived from black soldier fly larvae (BSFL) as a replacement for soybean oil in broiler chickens’ diets did not affect growth rates, feed preferences, blood characteristics, carcass features, or meat quality. Regardless of the inclusion of BSFL, different pieces of broiler chicken breast meat had similar levels of crude protein and fat content and showed comparable thawing loss. In addition, pH, color values, and drip loss were not affected by dietary treatments during refrigerated storage on day 0 and day 9. The fatty acid composition of broiler chicken breasts was significantly influenced by the amount of BSFL included in the diet. As the inclusion rate of BSFL increased, the proportion of saturated fatty acids increased, while the polyunsaturated fatty acid fraction decreased. On the other hand, the monounsaturated fatty acid fraction remained unchanged. The conclusion was reached that the inclusion of BSFL ensured satisfactory productivity, carcass characteristics, and overall meat quality. Therefore, BSFL has the potential to be a promising new chicken feed ingredient.

Józefiak et al. [72] assessed the impact of full-fat insect meal on performance and microbiota composition in the gastrointestinal tracts of broiler chickens. The trial incorporated *Gryllodes sigillatus*, *Shelfordella lateralis*, *Gryllus assimilis*, *T. molitor*, and *H. illucens* in variable proportions, ranging from 0.05% to 0.2%. This study found that the inclusion of insect meal in the broilers’ diets had no effect on their growth performance, but an increase in feed intake was observed. The inclusion of insect feeds lowered the pH levels of the digesta in both the crop and caeca. The introduction of *H. illucens* as a supplement had the greatest impact on the microbiota populations in the crop, ileum, and caeca. However, as the amount of *S. lateralis* added to the broiler diets increased, the count of selected microbiota in the crop and ileum also increased. These results show that using insect full-fat meals in small quantities can alter the microbiota composition in the digestive tract of broiler chickens.

Ducks serve as a viable and unique option for livestock. Kovitvadhi et al. [73] conducted a study to identify insect species that could serve as potential protein sources in the duck diet via in vitro digestibility tests. Yellow mealworm larvae, giant mealworm larvae (*Zophobas morio*), lesser wax moth larvae (*Achroia grisella*), house fly larvae, mulberry silkworm pupae (*Bombyx mori*), and American cockroach nymphs (*Periplaneta americana*) demonstrated high digestibility and were thus deemed excellent alternative protein sources for ducks. However, further research is necessary to confirm the findings of this study on incorporating insects into the diets of ducks at varying proportions and to bolster sustainable duck farming.

Elahi et al. [74] provided an informative summary of insect meal as a potential feed ingredient for poultry, citing 133 references that cover the literature on various insect species such as BSF, mealworms, houseflies, silkworms, termites (*Macrotermes subhyalinus, Macrotermes bellicosus, Glyptotermes montanus*), the cricket/grasshopper/locust group, and bees, as well as earthworms, which appears to be an outlier in this context. They concluded that insect meal (BSFM, mealworm meal, and housefly meal) presents the most promising prospects for industrialization in broiler and laying hen diets. However, earthworms, silkworms, and locusts can also be effectively utilized in poultry feed. Given the centuries-long use of insects in medicine, it is reasonable to consider replacing antibiotics used in poultry diets with insects due to their antimicrobial properties. Insect meal can be utilized in low-protein diets for amino acid adjustment due to insects’ high essential amino acid content.

Chickens naturally consume insects, which is why including insect products in their industrial feeds seems like a logical choice. However, it is important to educate consumers on the use of insects as a feed source to address fears and misconceptions that may arise. This objective was achieved by studies like Khalifah et al.’s [75], which explores the composition and nutritional benefits of insects, as well as their contribution to maintaining poultry farming and minimizing ecological hazards. Their analysis also assesses the pros and cons of insects and the potential of utilizing them as a primary nutrient source in upcoming years. Sajid et al.’s [76] review has similar content but that is obtained from different sources. They reference Raju et al.’s [77] paper, which summarizes two experiments on broiler chickens. The chickens were fed different levels (0, 0.25, 5, and 7.5%) of black soldier fly larva meal (BSFLM) from three sources, which had varying amounts of CP and CF. They discovered that bodyweight gain and food consumption increased consistently in the groups that were given BSFLM at up to 5.0% during the 0–3 weeks of age and remained similar throughout the rest of the period. However, the feed conversion rate (FCR) increased consistently over 3 to 5 weeks. When higher BSFLM levels (≥7.5% in the diet) were used in the second experiment, the body weight gain and food intake decreased significantly, and the FCR increased. Furthermore, dressing yield, breast weight, and abdominal fat content increased linearly with the level of BSFLM in the diet. It was found that the inclusion of BSFLM in broiler chicken feed at up to 5% can have positive impacts on growth during the early stages.

## 6. Use of Insect Meal in Feeding of Aquatic Animals

In his book chapter, Riddick [78] clearly summarized the then-contemporary situation of using insects as protein sources in aquaculture. He reviewed the research on four key species—the BSF, common house fly, silkworm moth, and yellow mealworm—that served as model insects to highlight progress. The protein and fat content differences between these species are important considerations as they vary between species and developmental stages within a species. The main conclusions of his research were that insects in the form of meal or pellets could provide sufficient protein to partially replace standard fish meal in the diets of fish that are omnivorous, such as catfish and carp, rather than carnivorous (trout and salmon). There is a need to develop cost-effective, large-scale farming practices to meet the increasing demand for farmed fish.

According to Freccia et al. [79], aquafeeds are primarily composed of cereals, oilseeds, and marine-origin components. The demand for feedstuff from the terrestrial animal industries poses a challenge to the profitability of aquafeeds, necessitating the identification of complementary ingredients. Many studies have focused on alternative protein sources, but research on plant proteins, microorganism-based proteins, and various animal by-products is ongoing to address constraints such as antinutritional factors and unbalanced nutrient profiles. The use of insects as a potential alternative in the nutrition of aquatic animals is still being investigated. Researchers are examining the advantages of utilizing insects as animal feed, the nutritional characteristics of various insect-based diets, advancements in aquatic feed technology, and considerations regarding the obstacles and future prospects of insect usage.

Insects have started to serve a significant function in aquaculture as substitute sources of protein. Insects are abundant in most freshwater environments, whereas in seawater, only three genera exist, with one occurring in the open sea and two in coral reef and tide pool marine environments. This group constitutes a natural part of the diets of both carnivorous and omnivorous fish [80] as well as other farm animals. Specifically, during the larval and fingerling stages of fish rearing, various insect species are crucial components of fish diets. Insects provide a valuable source of protein for fish in their natural habitat due to their high protein content ranging from 9.3% to 76% [48,51] and fat content ranging from 7.9% to 40% [81,82]. These variations lead to differences in the content of fatty acids and amino acids that need to be taken into account [83]. It should also be considered that insects are not only meal replacements but also prebiotics due to the presence of chitin and AMPs, so the inclusion of insect meal in fish diets, even at relatively low levels, could improve the immune system of fish and enhance their performance, as previously shown in other livestock species. However, it is also important to remember that there are more than 200 species of farmed fish, and their dietary requirements are not well understood. In addition, the process of the elaboration of insect meal before its use in aquatic animal feed should be considered [83].

## 7. Conclusions

In today’s world, with the increasing emphasis on sustainability and environmental awareness and with food production constituting a critical concern for a growing global population, a reassessment of the importance of insects is needed. Their ecological services are essential for sustainable agriculture, and edible insects offer a promising source of animal protein because they use land and water efficiently while recycling waste into nutrient-rich ingredients.

Insects are now being incorporated into feeding programs for land and aquatic animals due to their highly nutritious composition and availability. In addition, insect-derived components such as chitin, proteins, and fatty acids (with antimicrobial properties) offer other benefits. In the near future, there may be compelling economic reasons to replace other animal protein sources, particularly fishmeal, with insect-based feeds.

Low production volumes and high production costs render insect protein uncompetitive in terms of price compared to fishmeal or soybean meal. Nonetheless, this sector is witnessing notable growth concerning environmental and sustainability apprehensions, backed by comprehensive worldwide research efforts. Market statistics and forecasts reveal that this sector is expected to achieve a compound annual growth rate (CAGR) of 28.3% between 2022 and 2030.

In summary, the reviewed literature indicates that expanding insect production and utilization for food and feed offers the prospect of significantly bolstering agricultural sustainability, provided that ongoing extensive research is conducted to bridge knowledge gaps within this emerging field.

## Figures and Tables

**Figure 1 animals-14-01009-f001:**
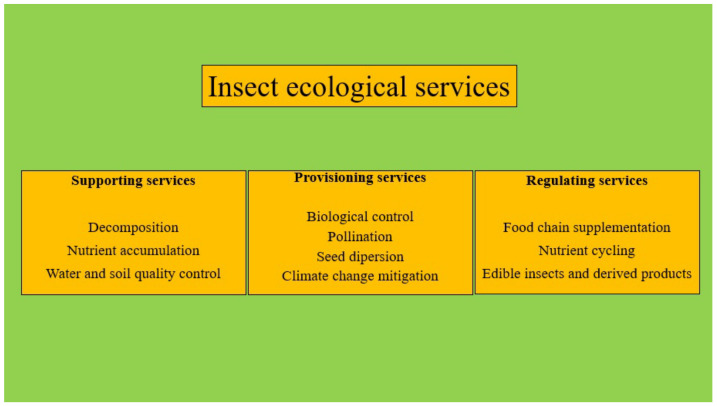
Ecological service categories for insects. Sources: [4,10].

**Table 2 animals-14-01009-t002:** The crude protein (CP) and crude fat (CF) content of some important edible insect species (dry weight, %).

Scientific Name	Common Name	CP	CF	References
*Orthoptera*	Crickets, locusts, grasshoppers (adult) house crickets grasshoppers	61.3 66.1 63.3 ± 5.7 50.5	13.4 21.9 17.3 ± 6.3 15.3	[46] [47] [48] [49]
*Tenebrio molitor*	Mealworm larvae defatted meal	47.1–49.4 58.4 52.8 ± 4.2 55.8 56.5	35.2–38.1 41.9 36.1 ± 4.1 25.2 6.2	[46] [47] [48] [49] [50]
*Zophobas morio*	Superworm	46.8	42.0	[47]
*Bombyx mori*	Silkworm spent pupae larvae pupae meal	58.0 53.8–69.8 60.7 ± 7.0	35.0 8.1–9.5 25.7 ± 9.0	[46] [48]
*Galleria mellonella*	Wax worm	34.0	60.0	[47]
*Musca domestica*	Common housefly larvae pupae	64.0 63.1 50.4	24.3 15.5 18.9	[46] [48] [48]
*Hermenia illucens*	Black soldier fly (BSF) defatted meal	45.1 56.1 52.5	36.1 25.2 9.3	[47] [49] [50]

## Data Availability

No new data were created or analyzed in this study. Data sharing is not applicable to this article.

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
