# Peer review of "The Role of Insects in Sustainable Animal Feed Production for Environmentally Friendly Agriculture: A Review"

_animals, 2024, doi:10.3390/ani14071009_

Round 1

Reviewer 1 Report

Comments and Suggestions for Authors

Review:  The role of insects in a sustainable animal feed production for 2 environmentally friendly agriculture: A review

We have seen many reviews on this topic.

I think this review is very simple and did not bring new references, any novelty or new conclusions

A review about insect meal nowadays should go deeper in nutrition, or presenting new potentials of use or calculations of environmental and sustainability parameters

An example, this review was submitted to a section of Aquatic Animals.

I know we have facing the big potential of feeding insect product on aquaculture, but the authors discussed it in 1 page

Table 1 is very complete in the original publications

Minor:

Keywords: insect; ecosystem service; insect product

L433 – italicize S. lateralis

L461 - Khalifah et al.

L 497 - Raju et al. please correct  and review others

L398 – De Marco et al.

L476 - BSFLM

After showing them for the first time you can abbreviate T. molitor and H. illucens

i.e. L392 is abbreviated and L399 is not.                                                   

L403 has an abbreviation Tenebrio molitor meal (TM) or Hermetia illucens meal (HI)

Please standardize it

Comments on the Quality of English Language

English is ok, but the scientific sound can improve

Reviewer 2 Report

Comments and Suggestions for Authors

List of observations from the evaluation of the article entitled . The role of insects in a sustainable animal feed production for environmentally friendly agriculture: A review. Manuscript I.D. 2896278

Abstract

Line 20 The chemical composition and nutritional value presented for insects is limited, e.g. vitamins, minerals, fatty acids etc. are not mentioned in tables.

Line 40 Explain for protein efficiency? The information reported refers to the protein concentration of some conventional foods and crickets.

Lines 48 and 50 Explain what is the difference between 0rganic matter recycling .... and to nutrient cycling.

Line 55 I suggest eliminating aquatic insects and write some linking words like these hexapods.

 Line 66 In Figure 1 I suggest deleting cultural services, since it does not match the title and both sources refer to ecology.

 Line 89 Write FAO [14].

Line 138 Clarify since an agroecological production system and an ecosystem are different.

 Line 151 Agrochemicals or fertlizantes????.

 Line 170 Write production techniques under controlled conditions, e.g. temperature and humidity.

Line 174 and 175 In all scientific names I suggest writing the name of the author and the year in which the corresponding description was made, for example Tenebrio molitor, Linnaeus, 1758.

Line 177 The information pets, zoo animals and animal production, is repeated on line 386.

Line 192 Table 1 In the first column I suggest writing Scientific Name, instead of species.

 Line 198 Clarify that in the larval stage of development.

Line 227 Insects also possess vitamins, amino acids, minerals, fatty acids, etc. This should also be clarified in lines 247 and 248, and also indicate for example crude protein and can also be quantified, true protein, digestible protein, amino acids, etc.

Line 231 Table 2 ,I suggest to eliminate Species or taxon and write Scientific Name, eliminate Orthoptera and write the scientific names of crickets, locusts, grasshoppers etc, as they did in all the other examples. Likewise in the common name column, write the developmental stage reported for crickets, locusts, superworm etc.

It is also necessary to revise the crude protein and crude fat concentrations, for example, in mealworm larvae 58.4 plus 41.9 add up to 103.3 which is illogical since in a proximate chemical analysis on dry basis the amounts of crude protein, ash, crude fat, crude fiber and carbohydrate or nitrogen free extract should be 100, it is also necessary to correct the genus Hermetia.

Line 272 Several authors have pointed out the production of biodiesel from insects and to reinforce this aspect I suggest writing examples of their production.

 Line 285 The word substrate refers to the feed or the diet used????

Line 304 Indicate that the data reported refer to dry basis.

Line 306 Suggest to report the average data 9.5 times

 Line 453 Clarify that earthworms are not insects.

References

I recommend revising the citation, since for example in reference 20 the name of the journal is misspelled and in reference 53 I suggest eliminating 0(0) since these numbers do not appear in the original citation and also that the citation be in accordance with the editorial norms of the journal.

Reviewer 3 Report

Comments and Suggestions for Authors

Your paper offers a comprehensive examination of the use of edible insects in sustainable agriculture, effectively highlighting their ecological and nutritional benefits compared to traditional livestock. You underscore the importance of sustainable practices in food production amidst growing population pressures and finite land resources. While the environmental and economic challenges of insect protein production are acknowledged, the focus predominantly rests on the nutritional advantages.

This reviewer suggests a greater emphasis on the economic hurdles currently facing the viability of insect protein production. It would be beneficial to either more closely align the conclusion with these economic considerations or adjust the main argument to fully reflect the environmental benefits discussed. This adjustment could provide a more balanced view of both the opportunities and challenges in utilizing insects as a sustainable food source.

Minor comments:

Please cite the source for the data mentioned on LL.40-42 regarding protein efficiency.

Reviewer 4 Report

Comments and Suggestions for Authors

This article analyzes the use of foods derived from insects in both terrestrial and aquatic farm animals, reviewing the existing information on productive parameters. The article provides an overview of the role of edible insects in the formulation of sustainable and more environmentally friendly feeds, which also provide added value as antimicrobials or prebiotics.

Although other reviews have recently been published on the use of insect meals and oils for feed formulation, this article offers a vision of sustainable and environmentally friendly production.

Some aspects should be considered:

The added value that insects can provide (antimicrobial and prebiotic) could be discussed in more depth.

Minor points:

Table 2 could be included entirely on page 7

Lines 303 to 306: could be rewritten, the idea that the authors want to give is not understood.

Line 319: “in concentration” could be deleted

Lines 116 to 117 and 130 to 131, talk about expectations in 2022, should be updated.

The sentences in lines 410 to 413 are repetitive, they could be rewritten.

Statements on lines 456 to 457 and lines 511 to 514 should be accompanied by references.

Line 511: prebiotics instead of probiotics

Line 517: aquatic animal instead of animal

Comments on the Quality of English Language

Line 89: “use” could be replaced by “the use”

Lines 194 and 195: “is” should be replaced by “are” and “aspect” by “aspects”

Line 282: The growth instead of growth

Round 2

Reviewer 1 Report

Comments and Suggestions for Authors

The manuscript still has the same problems mentioned in my R1

Comments on the Quality of English Language

Minor problems